# Total wash elimination for solid phase peptide synthesis

Jonathan M. Collins [1] ✉, Sandeep K. Singh [1], Travis A. White[1], Drew J. Cesta [1], Colin L. Simpson [1], Levi J. Tubb [1] & Christopher L. Houser[1]

We present a process for solid phase peptide synthesis (SPPS) that completely eliminates all solvent intensive washing steps during each amino acid addition cycle. A key breakthrough is the removal of a volatile Fmoc deprotection base through bulk evaporation at elevated temperature while preventing condensation on the vessel surfaces with a directed headspace gas flushing. This process was demonstrated at both research and production scales without any impact on product quality and when applied to a variety of challenging sequences (up to 89 amino acids in length). The overall result is an extremely fast, high purity, scalable process with a massive waste reduction (up to 95%) while only requiring 10–15% of the standard amount of base used. This transformation of SPPS represents a step-change in peptide manufacturing process efficiency, and should encourage expanded access to peptide-based therapeutics.

Peptide therapeutics are a unique and growing class of pharmaceuticals with high potency and selectivity for biological targets. Their use has grown dramatically with more than 80 peptide drugs approved by the FDA and hundreds in preclinical studies and clinical development[1,2]. As drugs, peptides have found application in a broad range of areas including cancer, metabolism, respiratory, cardiovascular, urology, autoimmune, pain, and antimicrobial applications[3–5]. In the last decade, a major success for peptide drugs has been the commercial approvals of peptide agonists of the glucagon like peptide-1 (GLP-1) receptor for type II diabetes treatment[6]. This includes drugs on the market such as liraglutide (Victoza®; Novo Nordisk), semaglutide (Ozempic®, Rybelsus®; Novo Nordisk), and dulaglutide (Trulicity®; Eli Lilly). More recently, semaglutide (Wegovy®; Novo Nordisk) has been approved for weight loss by the FDA which has dramatically increased the demand for its use[7].

Production of peptides can be performed through either chemical synthesis or biological methods such as recombinant deoxyribonucleic acid (rDNA) expression and fermentation[8]. Chemical synthesis is currently the standard method used for peptide production with its relatively fast production time, ability to incorporate non-standard derivatives, and lower risk of contamination with endotoxins. However, biological techniques have found use with longer natural sequences typically > 40 amino acids that can be more difficult to synthesize chemically. As notable examples, liraglutide was originally produced by rDNA and semaglutide by rDNA followed by a synthetic step. With growing interest in chemical synthesis, the FDA issued guidance on converting established rDNA production processes for GLP-1 analogues to a fully synthetic process with an abbreviated new drug application (ANDA)[9].

Since its inception in 1963, solid phase peptide synthesis (SPPS) has been a major enabling tool for peptide synthesis[10,11]. SPPS dramatically simplified the synthetic production of peptides compared to liquid phase peptide synthesis (LPPS) by allowing straightforward isolation of products with simple filtration at each step as opposed to more tedious extraction processes. Its use has been further enabled through improvements in resin technology[12], coupling reagents[13], and increased affordability of high-purity Fmoc amino acids. This has made the use of SPPS a fundamental and widely used technology for drug discovery and development toward a wide range of diseases. However, a major downside of SPPS is the significant waste it generates from successive washing steps between each deprotection and coupling step. Historically, about 5 washes have been needed between each step resulting in a large majority of the total waste generated and driving a high process mass intensity (PMI)[14]. With this limitation, improving the efficiency of SPPS was identified in 2016 by the ACS Green Chemistry Institute Pharmaceutical Roundtable as a critical unmet need[15].

[1]Peptide Synthesis Research, CEM Corporation, 3100 Smith Farm Rd, Matthews, NC 28104, USA. ✉e-mail: jon.collins@cem.com

The use of microwave energy and heating, in general, was initially applied to SPPS for accelerating synthesis times and improving purity by driving reaction steps toward completion[16–18]. Later developments reduced the need for washing after each coupling step[19–22], but up to this point, washing after the deprotection step has been unavoidable. Typically, if residual base from deprotection contaminates the next coupling step, it will remove the Fmoc protecting group on the next amino acid leading to the undesirable insertion of an additional amino acid onto the growing chain. Furthermore, the residual base can react with and consume activated amino ester before it reacts with the peptide terminus. The result is the generation of both insertions and deletions of the next amino acid, which can lead to total failure of the synthesis. Post-deprotection washings have been indispensable and consume the largest amount of solvent in solid-phase peptide synthesis process with approximately 90% of waste generated by these washings.

With the goal to remove the need for washing, we started by considering alternative processes to eliminate the residual deprotection base. Options to quench the excess base are not straightforward since the deprotected peptide chain also has a free amine which would likely be capped in the process. Furthermore, the additional cost of introducing a specialized quenching reagent for this approach was undesirable. Removal by extraction as used in LPPS appeared similarly undesirable due to difficulty automating, use of extra solvent for phase separation, and potential yield loss at each step. In contrast, an evaporative-based process for base removal appeared uniquely suitable for microwave SPPS. Under microwave SPPS conditions, the deprotection reaction is at an elevated temperature and the boiling point of the amine bases (piperidine lit.[23] 106 °C, pyrrolidine lit.[23] 87 °C) are less than the standard solvents N,N-dimethylformamide (DMF lit.[23] 153 °C) or N-methyl-2-pyrrolidone (NMP lit.[23] 202 °C). Additionally, nitrogen bubbling used for mixing during the deprotection step encourages evaporation. We, therefore, investigated whether we could utilize evaporation as a process to gradually remove excess base from the deprotection step.

While piperidine is the standard deprotection base used, pyrrolidine with a similar $pK_a$ has been identified as an alternative base[24,25]. Its smaller 5-membered ring is attractive for potentially accelerating Fmoc removal vs. piperidine. Additionally, pyrrolidine has a lower boiling point than piperidine (87 °C vs. 106 °C) and has been previously utilized in our lab at temperatures significantly above this during the deprotection step. We, therefore, envisioned pyrrolidine as advantageous in place of piperidine for both allowing a lower excess of base and being easier to remove by evaporation.

With proper conditions, a continual time-based decrease in the amount of pyrrolidine could allow complete deprotection and scavenging of the Fmoc group, and after a certain amount of time would reach an adequately low concentration such that it does not impact the next coupling.

Evaporation of base in a batch SPPS reaction vessel requires consideration of the cleanliness of the vessel surfaces. Base condensation can occur anywhere on the upper parts of the reaction vessel and may be undesirably introduced into subsequent coupling steps as droplets that fall back into the solution. Therefore, achieving a goal of no washing after the deprotection step requires not only the removal of base from the solution but also preventing condensation on the upper reaction vessel surfaces that can lead to subsequent re-introduction into the system. Nitrogen bubbling during the deprotection step provides gas flow for moving vapors out of the reaction vessel headspace and out of the vent line. However, the upward direction of nitrogen from bubbling can encourage splashing without a directional force to return any resulting droplets to the solution. We therefore introduced a second source of $N_2$ through a dedicated line into the headspace above the reaction vessel with an exit through a vent port (Supplementary Fig. 1). This results in both a higher gas exchange rate above the deprotection solution and a top-down directional flow which pushes condensation back into the reaction vessel solution where it is reheated and re-evaporated. We used a one-pot deprotection-coupling methodology[20,21] where an optimized amount of deprotection base could be added directly to the post-coupling solution without any draining, this allows for the reuse of both solvent and heat from the coupling solution to facilitate the deprotection process resulting in further solvent and time savings. The concept of utilizing bulk evaporation with flushing of the reaction headspace to remove the residual pyrrolidine combined with active ester quenching after the coupling step to achieve wash-free SPPS is represented in Fig. 1 [26].

## Results and Discussion
### Background
The methodology was designed using traditional carbodiimide based activation with N,N′-diisopropylcarbodiimide (DIC) and ethyl 2-cyano-2-(hydroxyimino)acetate (Oxyma Pure). Unlike phosphonium and aminium-based coupling, this activation strategy is extraordinarily tolerant to elevated temperatures because it avoids the use of excess base, such as DIEA, which leads to epimerization and other problematic side reactions when heated[19]. At the end of the coupling reaction, components in the reaction mixture include Fmoc-protected

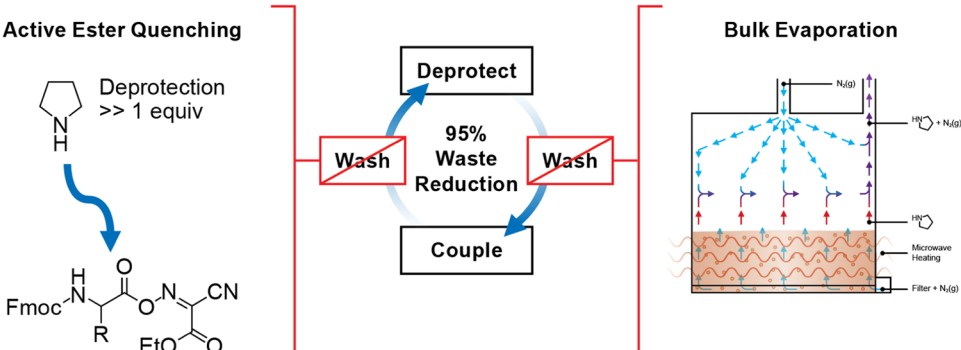

# Wash-Free SPPS

**Active Ester Quenching** **Bulk Evaporation**

**Fig. 1 | A Wash-Free SPPS Process.** One-pot coupling-deprotection methodology involves the addition of an optimized amount of deprotection reagent (pyrrolidine) to the undrained post-coupling mixture at elevated temperature resulting in rapid quenching of the active ester. Bulk evaporation (purple arrows) of pyrrolidine (red arrows) is accomplished via microwave heating. Subsequent removal of pyrrolidine vapor from the vessel is achieved by headspace gas flushing with $N_2$ (blue arrows). The combination of one-pot coupling-deprotection methodology and reaction vessel headspace flushing allows the elimination of all wash steps resulting in 95% waste reduction in SPPS.

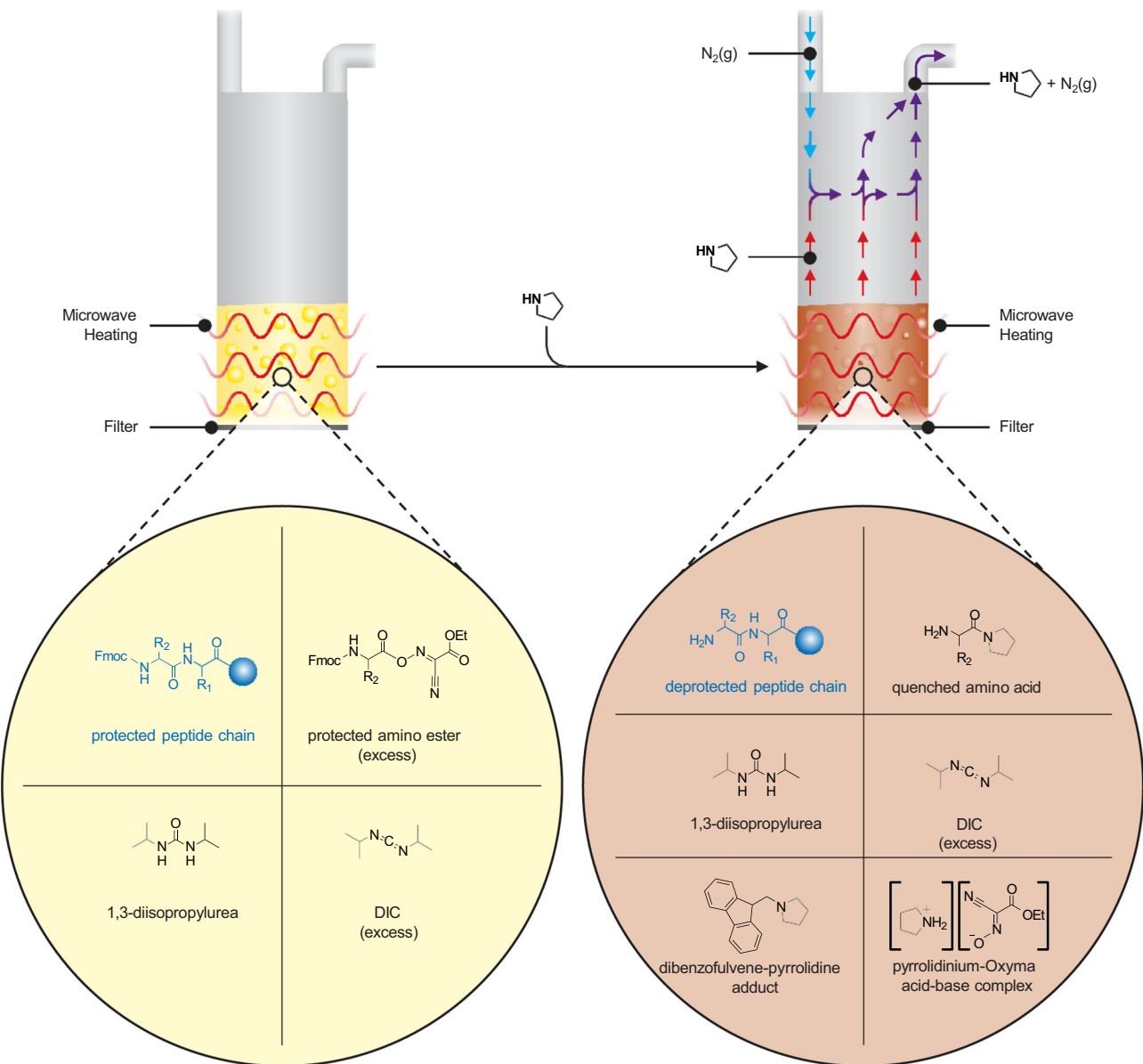

**Fig. 2 | Reaction products during wash-free bulk-evaporation SPPS utilizing N₂ as the headspace flushing gas.** Reaction mixture in the coupling step includes Fmoc-protected peptide resin, amino ester, 1,3-diisopropylurea and DIC. At the end of the coupling step, the introduction of pyrrolidine results in (i) hydrolysis of the leftover amino ester resulting in the formation of quenched amino acid and pyrrolidinium-Oxyma acid-base complex, and (ii) Fmoc-deprotected peptide resin and dibenzofulvene-pyrrolidine adduct. Headspace gas flushing with N₂ removes pyrrolidine vapors during the deprotection step.

peptide resin, unreacted excess Fmoc-amino ester, leftover DIC and 1,3-diisopropyl urea (Fig. 2). At this stage, pyrrolidine was added directly to the post-coupling mixture to quench the excess coupling reagents and begin the Fmoc-deprotection of peptide resin. At the end of the deprotection reaction, the quenched species including pyrrolidine-capped amino acid, pyrrolidinium-Oxyma complex and dibenzofulvene-pyrrolidine adduct are then drained from the reaction along with the remaining pyrrolidine. Microwave heating was applied throughout the coupling and deprotection reactions to increase the speed of the reactions and encourage the evaporation of pyrrolidine. The elimination of pyrrolidine vapors from the reaction vessel was assisted by flushing the headspace of the vessel with N₂ throughout the deprotection reaction.

## Establishment of baseline conditions

For initial development of the process parameters, the Jung-Redmann (JR) peptide was chosen as a model sequence. This peptide has a difficult sequence that is known to aggregate during SPPS causing slowed kinetics for Fmoc deprotection[27–30] and coupling[31]. It provides a means to determine if the deprotection method will provide robust Fmoc removal. We first decreased the amount of base added to the deprotection method in order to minimize the energy needed for complete evaporation. Standard deprotection solutions with 20% piperidine contain a large excess of base that is typically 10–20 fold larger than the excess of amino acid used. Here, the use of elevated temperatures (80–110 °C) allows the deprotection reaction to complete rapidly with less than 5% pyrrolidine. It was also recognized that a moderate substitution (0.2–0.3 mmol/g) typically utilized on either polyethylene glycol-polystyrene (PEG-PS) or lower-loading polystyrene (PS) resins could facilitate the deprotection reaction to support the use of a lower amount of base. Since Oxyma is regenerated after acylation, we recognized that one less equivalent of Oxyma could be used during the coupling reaction. This deficit of Oxyma (pka 4.60)[32] would reduce the acidity of the coupling mixture and therefore

**Table 1 | Optimization of wash-free conditions on the JR sequence (WFTTLISTIM-NH$_2$) with 110 °C Fmoc removal**

| Entry | Deprotection (% pyrrolidine) | Deprotection Time (sec) | Post-Deprotection Washing | Headspace Flushing | Crude Purity[5] (%) |
|---|---|---|---|---|---|
| 1 | 4.5 | 40 | 2 × 4 mL DMF | OFF | 77 |
| 2 | 4.5 | 40 | 2 × 4 mL DMF | ON | 79 |
| 3 | 4.5 | 40 | None | OFF | 16 |
| 4 | 4.5 | 40 | None | ON | 69 |
| 5 | 4.5 | 80 | None | OFF | 20 |
| 6 | 4.5 | 80 | None | ON | 68 |
| 7 | 3.5 | 80 | None | OFF | 34 |
| 8 | 3.5 | 80 | None | ON | 86 |
| 9 | 3 | 80 | None | OFF | 56 |
| 10 | 3 | 80 | None | ON | 84 |
| 11[1] | 3 | 80 | None | ON | 73 |
| 12 | 3 | 40 | None | ON | 73 |
| 13 | 2 | 80 | None | OFF | 36 |
| 14 | 2 | 80 | None | ON | 65 |
| 15 | 3 | 80 | 2 × 4 mL DMF | ON | 81 |
| 16[2] | 4.5 | 40 | 2 × 4 mL NBP | ON | 74 |
| 17[2] | 3 | 80 | None | ON | 65 |
| 18[3] | 4.5 | 40 | 2 x 4 mL NMP | ON | 70 |
| 19[3] | 3 | 80 | None | ON | 66 |
| 20[4] | 4.5 | 40 | 2 × 4 mL DMF | ON | 72 |
| 21[4] | 4.5 | 40 | None | ON | 65 |
| 22[4] | 3 | 80 | None | ON | 75 |

Notes: 1. Fmoc-Rink Amide ProTide[TM] LL resin (0.20 meq/g substitution) was used for all experiments except entry 11 that used Fmoc-Rink Amide MBHA PS resin (0.33 meq/g substitution); **2.** N-butylpyrrolidinone (NBP) was used in place of DMF for entries 16 and 17. **3.** N-methylpyrrolidinone (NMP) was used in place of DMF for entries 18 and 19. **4.** Piperidine was used in place of pyrrolidine for entries 20-22. **5.** UPLC-MS chromatograms and purity reports have been provided in the Supplementary Information.

help minimize the amount of pyrrolidine needed to quench the coupling mixture. Furthermore, limiting free Oxyma during the coupling step is attractive for the protection of acid-sensitive side-reactions and potential HCN formation from reaction with DIC[33].

As shown in Table 1, we explored the synthesis of the JR sequence using variable amounts of pyrrolidine base under variable deprotection times and resins both with and without additional nitrogen headspace flushing, keeping coupling conditions the same for all experiments. Syntheses were performed with a Liberty PRIME 2.0 automated microwave peptide synthesizer at 0.1 mmol scale utilizing one-pot coupling and deprotection steps.

First, a control synthesis of the JR sequence was conducted using our standard one-pot conditions with 2 washes after deprotection (Table 1, entry 1). Applying directed headspace flushing did not significantly improve the purity with post-deprotection washing still being applied (Table 1, entry 2). Then, the synthesis was repeated without washing (Table 1, entry 3) resulting in a significant drop in purity due to incomplete removal of pyrrolidine from the coupling solution that caused the formation of deletion side products of Trp, Phe, Ile/Leu. When headspace flushing was used (Table 1, entry 4), the purity was substantially improved but still reduced versus the control. This led us to investigate increasing the deprotection reaction time to allow for further evaporation of pyrrolidine as we continued to decrease the pyrrolidine concentration (Table 1, entries 6, 8, 10, and 14). Both changes were beneficial with an ideal pyrrolidine concentration determined between 3–3.5% for 80 seconds reaction time (Table 1, entries 8 and 10). Headspace flushing provided a clear benefit when eliminating washing for each deprotection concentration tested (Fig. 3b). Figure 3a shows a typical chromatogram with low crude purity due to the formation of various side products involving insertions and deletions of Trp, Phe, Ile/Leu, and Met when the wash-free method was tested with headspace flushing turned off (Table 1, entries 5, 6). High crude purities were obtained for the

corresponding experiments when headspace flushing was turned on (Fig. 3b). No further increase in crude purity was observed by adding 2 ×4 mL post-deprotection washings to the fully optimized method (Table 1, entry 15). Thus, we were able to eliminate washing for the JR sequence without any significant drop in purity from the optimized control conditions when using 3–3.5% pyrrolidine for 80 seconds deprotection at 110 °C (Fig. 3c). Use of these optimized conditions for the wash-free process allowed high purity synthesis of JR with a complete cycle waste of only 4.25 mL per amino acid and a total cycle time of approximately 3.5 minutes.

In recent years, greener solvent replacements have been explored for SPPS[34–37]. We therefore attempted to understand if this new process could work with solvents other than DMF. For this wash-free process it is important to utilize solvents with significantly higher boiling points than the deprotection base to prevent the mixture from evaporating to dryness. Therefore, we evaluated N-methylpyrrolidinone (NMP) and N-butylpyrrolidinone (NBP), which have higher boiling points than DMF. NBP is attractive with its non-reprotoxic properties in comparison to both DMF and NMP[34]. With both NMP and NBP, similar results were obtained between the wash-free and wash-based controls (Table 1, entries 16-19). The use of NBP and NMP did show a slight reduction in purity compared to DMF, however, the syntheses were still successful in producing the target in relatively high purity with both the wash-based and wash-free conditions. While further optimization may be beneficial for the use of NBP and NMP, these initial results indicate that this new process works well not just with DMF, but also with alternative solvents.

Piperidine has been the most commonly used Fmoc deprotection base in traditional SPPS. Therefore, it was of interest to compare the results with the optimized conditions when piperidine was used in place of pyrrolidine. We started with a control synthesis of JR sequence using piperidine and our standard one-pot conditions with post-deprotection washings (Table 1, entry 20), which showed a slight

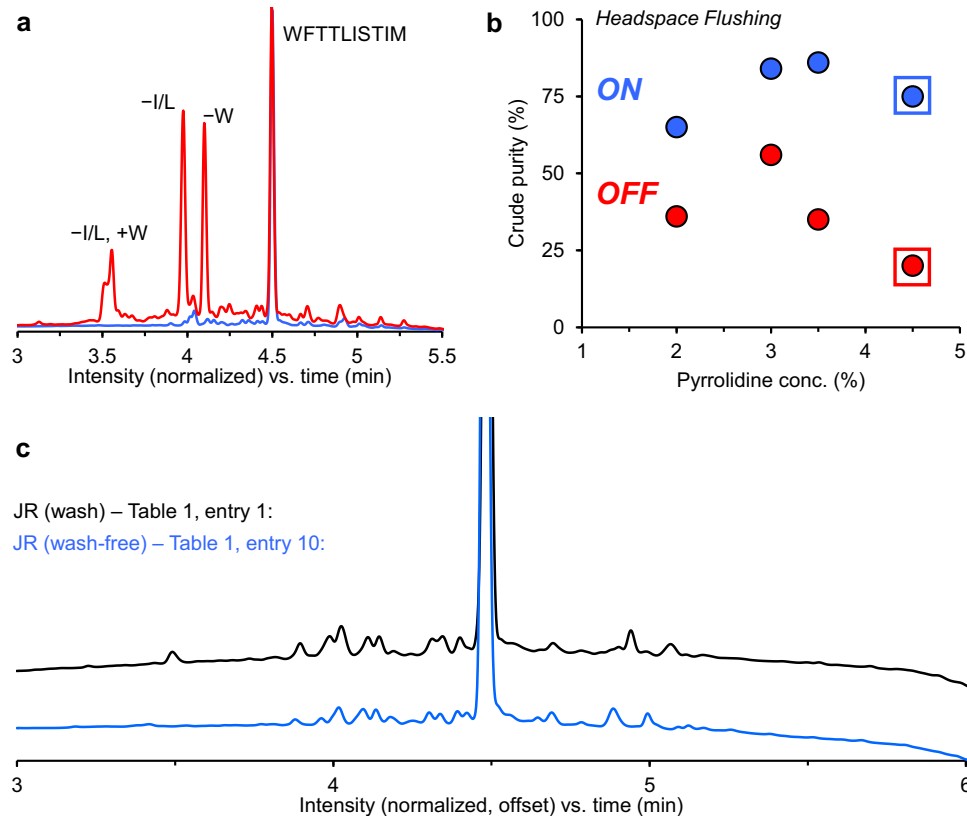

**Fig. 3 | Crude purity data analysis for the synthesis of JR. a** Crude chromatogram of JR showing typical insertion and deletion signals when headspace flushing is turned off in wash-free method. **b** Crude purities from wash-free optimization experiments with headspace flushing turned on (blue) and off (red). Crude purities were always higher for experiments with headspace flushing turned on and lower when headspace flushing was off for runs with identical deprotection (pyrrolidine) concentrations. The use of 3–3.5% pyrrolidine concentration showed the highest purity, using lower and higher concentrations resulted in purity drop. **c** Crude chromatograms of JR showing very similar purity profiles for the standard one-pot method with 2 ×4 mL post-deprotection washing and the wash-free optimized method. All chromatograms were obtained with UV detection at 214 nm.

reduction in crude purity when compared with the corresponding experiment using pyrrolidine (Table 1, entry 2). Since washing was used in both runs, the increase in deletion impurities indicates a slightly lower reactivity of piperidine as compared to pyrrolidine. Removing post-deprotection washings under these conditions showed a similar reduction in crude purity (Table 1, entry 21) as compared to the corresponding run with pyrrolidine (Table 1, entry 4). Finally, the fully optimized wash-free conditions were tested with piperidine and showed crude purity of 75% (Table 1, entry 22), while under identical conditions a crude purity of 84% was obtained with pyrrolidine (Table 1, entry 10). The additional impurities were identified as insertions and deletions of Trp, Phe, Ile/Leu, and Met. These results show that piperidine can be used under wash-free conditions, but may require further optimization to achieve the same crude purities as pyrrolidine due to the decreased reactivity and rate of evaporation.

**Validation of baseline conditions**

Satisfied by the JR synthesis results, we then applied this wash-free process to other well-known, difficult sequences using a 3% pyrrolidine concentration. The first was the [65-74]ACP (acyl carrier protein) test peptide that showed very similar crude purity for wash-based and wash-free methods (Supplementary Table 1, Fig. 4a). We then studied the commercially-available medications liraglutide and semaglutide, which are glucagon-like peptide-1 receptor agonists (GLP-1-RA) often prescribed for treating type 2 diabetes. Their synthesis has been described as difficult and previously required either synthesis in smaller fragments or utilizing multiple pseudoprolines for high purity

synthesis[38,39]. The third test sequence was the 42-residue fragment [1-42]β-amyloid peptide which has been identified in plaques formed with Alzheimer's disease. This peptide is notoriously difficult to synthesize and often requires the use of special reagents and extended synthesis conditions due to its hydrophobic nature[40–42]. The synthesis of liraglutide, semaglutide and [1-42]β-amyloid sequences was achieved with very similar crude purities between wash-based and the new wash-free process as shown in Supplementary Table 1 and Fig. 4a.

**Wash-free production scale synthesis**

In preparation for testing the wash-free method at a production scale, we first investigated the 25 mmol scale deprotection and coupling conditions by synthesizing liraglutide at a research scale (0.1 mmol) in a 35 mL reaction vessel; this method employed lower temperature conditions for deprotection (8 min at 80 °C) and coupling (5 min at 80 °C). Pleasingly, this method yielded liraglutide with crude purity that matched previous results obtained with wash-based methods at research and production scales.

Having established the chromatographic purities using large-scale reaction conditions for 0.1 mmol scale synthesis, we then proceeded to the 25 mmol wash-free synthesis of liraglutide on the Liberty PRO large-scale microwave peptide synthesizer (Fig. 5). Results from an initial optimization experiment involving 6 amino acid couplings suggested that the ideal conditions for a 25 mmol wash-free synthesis should be (i) 2.5% pyrrolidine concentration in the reaction vessel, (ii) 10 min at 90 °C deprotection and 5 min at 80 °C coupling, and (iii) 85 L/min nitrogen pressure of directed headspace flushing during each deprotection. We used a 4 equivalent excess of regular amino

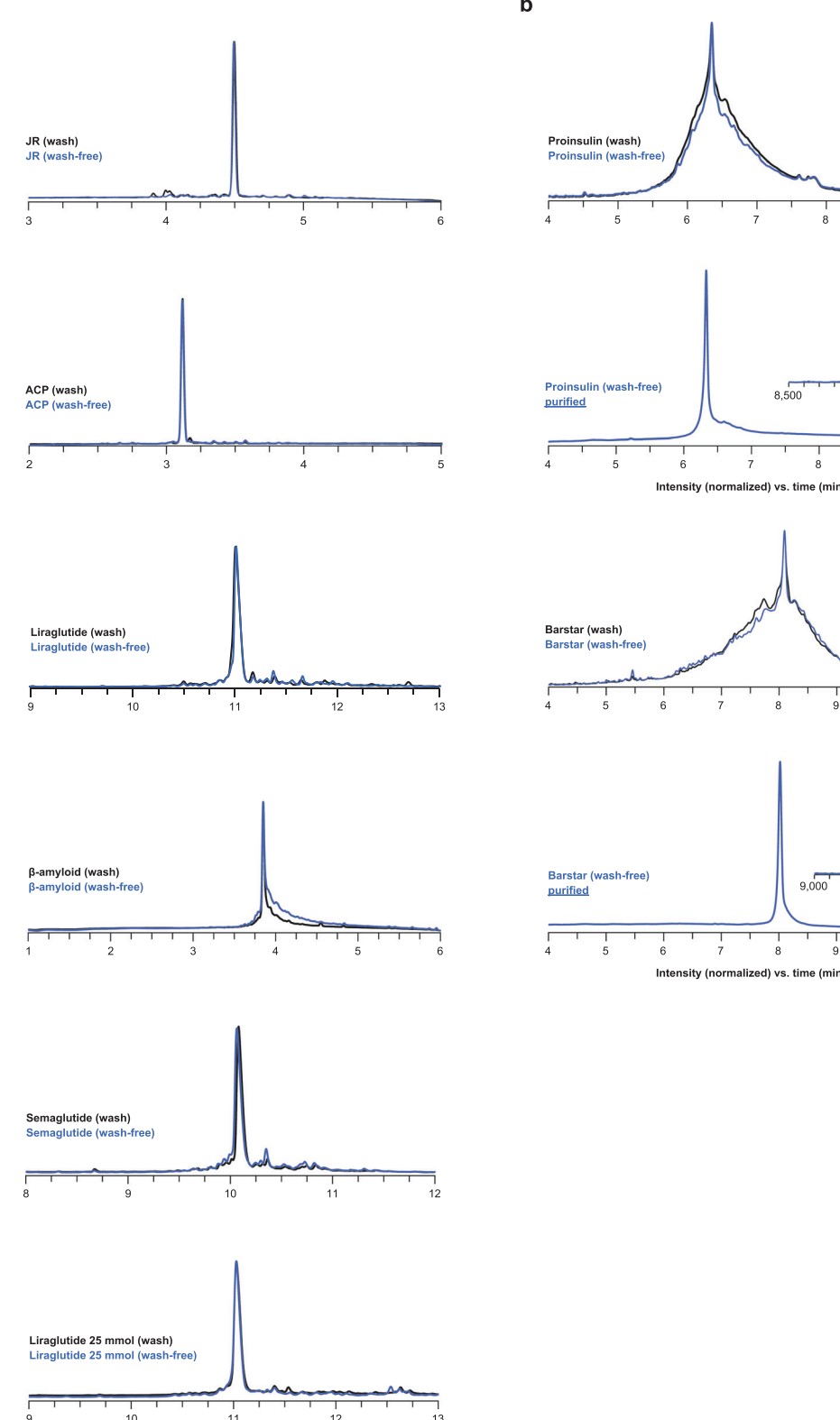

**Fig. 4 | Comparison of wash-free and wash-based crude purities. a** peptide synthesis. **b** protein synthesis. All chromatograms were obtained with UV detection at 214 nm. JR: WFTTLISTIM-NH$_2$; MW = 1211 Da; $^{65-74}$ACP: VQAAIDYING-OH; MW = 1063 Da; Liraglutide: HAEGT-FTSDVSSYLEGQAAK(γ-E-palmitoyl)-EFIAWLVRGRG-OH; MW = 3751 Da; Semaglutide: HXEGTF-TSDVSSYLEGNAAK(C18 diacid-γ-E-OEG-OEG)-EFIAWLVRGRG-OH, where X = Aib; MW = 4113 Da; $^{1-42}$β-amyloid: DAEFRHD SGYEVHHQKLVFFAEDVGSNKGAIIGLMVGGVVIA-OH; MW = 4514 Da; Proinsulin: FVNQHLCGSHLVEALYLVCGERGFFYTPKTRREAEDLQVGQVELGGGPGAGSLQPLA LEGSLQKRGIVEQCCT-SICSLYQLENYCN-NH$_2$; MW = 9394 Da; Barstar: KKAVING EQIRSISDLHQTLKKELALPEYYGENLDALWDCLTGWVEYPLVLEWRQFEQSKQLTEN GAESVLQVFR-EAKAEGCDITIILS-NH$_2$; MW = 10210 Da.

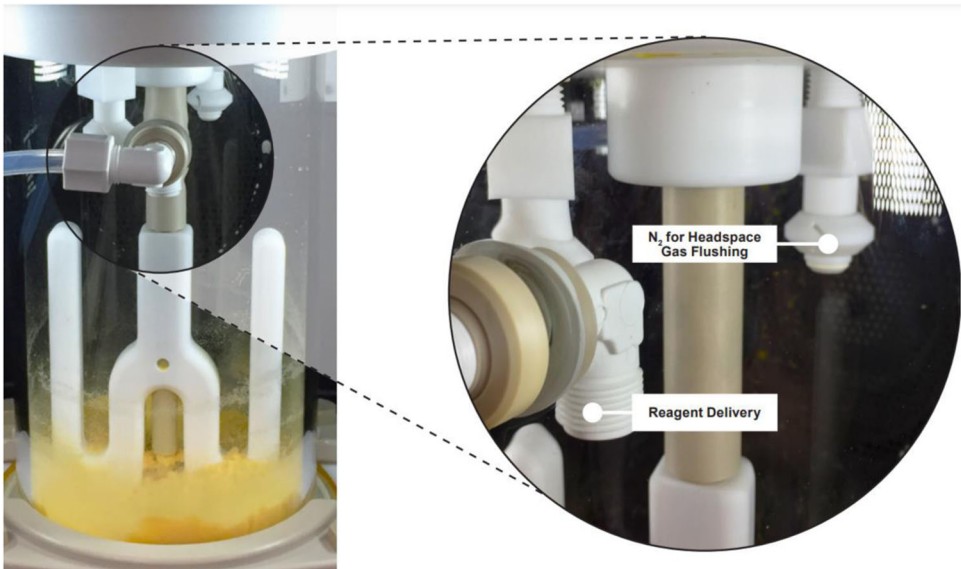

**Fig. 5 | 3 L Microwave reaction vessel used for 25 mmol production scale synthesis of liraglutide.** The borosilicate glass vessel utilizes combined mechanical stirring and nitrogen bubbling from underneath the frit for mixing. The expanded view highlights dedicated ports for N$_2$ flushing of the reaction headspace gas and additionally for reagent delivery.

acids and only 2 equivalents of Fmoc-Lys(palmitoyl-Glu-OtBu)-OH for coupling. Under these conditions, the 25 mmol wash-free synthesis of liraglutide generated a total waste of 28.4 L as compared to 139.7 L from the wash-based 25 mmol run which implies an overall waste reduction of approximately 80%. Liraglutide samples from wash-based and wash-free 25 mmol syntheses showed very similar (77–78%) crude purities (Supplementary Table 1, Fig. 4a). These results confirm the general applicability of wash-free methodology for research scale as well as production scale synthesis of peptides. Future use of this technique in large-scale peptide drug production would be helpful in reducing enormous amounts of waste generated from SPPS.

The potential for epimerization with this new wash-free process was evaluated by measuring the occurrence of D-amino acids in the liraglutide samples. Liraglutide samples synthesized by both the research and production scale wash-free methods (Supplementary Table 1, entries 2 and 5) were analyzed for epimerization using an established method involving hydrolysis, derivatization, and subsequent GC-MS analysis (CAT GmbH)[43]. The results from the crude and corresponding purified samples were then compared to a commercial sample of liraglutide (Victoza®) as shown in Table 2. Encouragingly, the results showed very low levels of epimerization with well over 99.5% control of stereochemistry for each amino acid in the liraglutide sequence using the production method. Importantly, this demonstrates that all epimerization related impurities are below the critical 0.5% limit for any new specified peptide-related impurity. Meeting this limit is required for the application of a synthetic peptide as a substitute for an approved peptide drug of rDNA origin with an ANDA[9]. Therefore, these results demonstrate the potential of this process to be used not only in R&D, but also production processes that require stringent purity standards.

## Application to protein synthesis

The capabilities of this process were then tested further by the synthesis of two proteins with sequence lengths >80 amino acids, proinsulin and barstar. Linear synthesis of long sequences by SPPS is challenging due to the iterative accumulation of impurities and increased susceptibility for aggregation to occur. The proinsulin 86-mer and barstar 89-mer sequences were chosen for synthesis as they were previously synthesized using a fast flow methodology at

1% and 2% overall yield, respectively[44]. The fast flow approach advantageously provided a very fast synthesis time of only ~2.5 minutes per amino acid cycle at small synthesis scales (0.035 mmol for proinsulin; 0.027 mmol for barstar). However, the process required a large excess of amino acid (~100 equivalents) and wash solvent (~90 mL per amino acid).

To account for the potential increased synthesis difficulty of these longer sequences, we utilized a higher coupling concentration with 10 equivalents of amino acids and extended the deprotection (2 minutes) and coupling times (4.5 minutes). The pyrrolidine concentration was also increased to 3.8% to quench the larger excess of activated amino acid. These conditions resulted in a cycle time of ~7.3 minutes per amino acid, and a total waste of 5.5 mL per amino acid at 0.1 mmol synthesis scale. Using this wash-free process, we obtained both proteins with similar crude purity as when washing was utilized (Fig. 4b). The crude proinsulin and barstar samples were then purified by reversed-phase HPLC which resulted in 2.4% and 3.4% overall yield, respectively. Purified samples of proteins were identified by deconvoluted mass spectra showing 9395 Da and 10210 Da for proinsulin and barstar, respectively (Fig. 4b). These protein synthesis examples demonstrate that the wash-free process is robust for generating high purity results even for long and challenging sequences (Supplementary Table 2).

## Quantification of residual pyrrolidine by GC-FID

To further validate the removal of pyrrolidine by this process, a study was performed using gas chromatography with flame ionization detection (GC-FID) to quantify the amount of residual pyrrolidine after completing a deprotection step, draining the mixture, and adding fresh DMF to the reaction vessel. Previously established wash-based SPPS processes reported residual piperidine concentrations on the order of 500–2000 ppm[24,45]. These trace amounts of deprotection base have minimal effect on the subsequent coupling reaction and are normally tolerated in wash-based SPPS with the understanding that additional washes will have diminishing returns on the quality of the peptide produced and create unnecessary waste. For the optimized wash-free conditions (Table 1, entry 10) with a starting pyrrolidine concentration at around 30,000 ppm (3% pyrrolidine), the GC-FID experiment (Supplementary Table 5) was performed and determined a

**Table 2 | Epimerization data for liraglutide samples**

| Residue | D-Enantiomer | | | | | |
|---|---|---|---|---|---|---|
| | Supplementary Table 1, Entry 2 (Crude) 0.1 mmol research method | Supplementary Table 1, Entry 2 (Purified) 0.1 mmol research method | Supplementary Table 1, Entry 5 (Crude) 0.1 mmol production method | Supplementary Table 1, Entry 5 (Purified) 0.1 mmol production method | Supplementary Table 1, Entry 6 (Purified) 25 mmol production run | VICTOZA® Lot #FS61B71 |
| Alanine | 0.15% | 0.10% | 0.12% | 0.10% | 0.10% | <0.10% |
| Valine | <0.10% | <0.10% | <0.10% | <0.10% | <0.10% | <0.10% |
| Threonine | <0.10% | <0.10% | <0.10% | <0.10% | <0.10% | <0.10% |
| | <0.10% D-allo | <0.10% D-allo | <0.10% D-allo | <0.10% D-allo | <0.10% D-allo | <0.10% D-allo |
| | <0.10% L-allo | <0.10% L-allo | <0.10% L-allo | <0.10% L-allo | <0.10% L-allo | <0.10% L-allo |
| Isoleucine | <0.10% | <0.10% | <0.10% | <0.10% | <0.10% | <0.10% |
| | 0.11% D-allo | <0.10% D-allo | <0.10% D-allo | <0.12% D-allo | <0.12% D-allo | <0.10% D-allo |
| | <0.10% L-allo | <0.10% L-allo | <0.10% L-allo | <0.13% L-allo | <0.14% L-allo | <0.11% L-allo |
| Leucine | 0.12% | 0.16% | 0.14% | 0.15% | 0.13% | 0.13% |
| Serine | 0.31% | 0.10% | 0.21% | <0.10% | <0.10% | 0.38% |
| Aspartic Acid | 0.36% | 0.38% | 0.17% | 0.33% | 0.18% | 0.12% |
| Phenylalanine | 0.19% | 0.22% | 0.15% | 0.22% | 0.14% | 0.13% |
| Glutamic Acid | 0.23% | 0.37% | 0.23% | 0.30% | 0.25% | 0.16% |
| Tyrosine | 0.15% | 0.10% | <0.10% | 0.14% | <0.10% | 0.18% |
| Lysine | 0.13% | <0.10% | 0.15% | 0.11% | 0.13% | 0.11% |
| Arginine | 0.23% | 0.11% | 0.24% | 0.16% | 0.10% | 0.13% |
| Tryptophan | 0.23% | <0.10% | 0.22% | 0.16% | not determined | 0.25% |
| Histidine | 0.72% | 0.72% | 0.40% | 0.34% | 0.47% | 0.57% |

residual pyrrolidine in the subsequent coupling solution at 1680 ppm with only an 80 second evaporation time. Further reductions in pyrrolidine appear possible by simply extending the evaporation time.

A completely wash-free process for solid phase synthesis has been demonstrated for the routine synthesis of peptides and even proteins near 100 amino acids at both research and production scales. Its use did not impact product purity versus controls with washing, and combined with its inherent elevated temperature reaction conditions provides high purity and rapid synthesis times. Compared to traditional SPPS, this new wash-free process provides up to a 95% reduction in waste generated whereas current manufacturing processes can require multiple deprotection steps and up to 10 washes per amino acid addition[45].

By completely eliminating wash solvent and reducing the amount of deprotection base, the wash-free process reduces raw material requirements and subsequent waste disposal which has important benefits for the production of peptide drugs. Implementation of the wash-free process into peptide production provides the opportunity to annually eliminate millions of liters of harmful solvents. This is based on the extensive and growing use of SPPS globally driven by approvals for peptide drugs such as linaclotide (Linzess; Ironwood/Forest), plecanatide (Trulance; Synergy), and semaglutide (Ozempic®, Rybelsus®, Wegovy®; Novo Nordisk) that have pushed manufacturing requirements to quantities routinely above 100 kg per year and even approaching metric ton quantities[46]. Correspondingly, waste from current peptide manufacturing processes is typically 3000–15000 kg per kg API with a majority of the waste composition from DMF[47]. Furthermore, the minimization of overall solvent and reagents needed can help overcome cost barriers to encourage the use of greener solvents and reagents that are highly desirable, but more expensive. This benefit will aid compliance with tighter restrictions soon taking effect for the use of DMF[48]. Finally, the use of optimized microwave-assisted reaction conditions results in a higher crude purity with a rapid synthesis time that improves quality while reducing purification costs[16–19]. Together, the fundamental advancements realized from the new wash-free process provide a pathway for SPPS to meet the sustainability needs of modern drug development and production.

## Methods
### Peptide synthesis
All peptides were synthesized using automated microwave synthesis conditions on a CEM Liberty PRIME 2.0 system at 0.1 mmol scale using the one-pot coupling/deprotection methodology[20,21]. Method details involving reaction time, temperature, and concentration of deprotection reagent are described in Table 1 and Supplementary Table 1. Couplings were performed for 30 s at room temperature followed by 60 s at 105 °C using Fmoc-amino acid (1.0 mL, 0.5 M in DMF), DIC (1.0 mL, 0.75 M in DMF) and Oxyma (1.5 mL, 0.26 M in DMF). Fmoc deprotection step was initiated by adding 0.75 mL of pyrrolidine/DMF (17% v/v) directly to the undrained post-coupling solution (optimization experiments were performed by adding 0.75 mL of 11.3–25% v/v pyrrolidine/DMF as described in Table 1). Nitrogen headspace flushing (3.5 L/min) was used during the deprotection step. The wash-based method used 2 ×4 mL DMF post-deprotection washings. The cycles involving deprotection-coupling (for wash-free) or deprotection-washing-coupling (for wash-based) runs were automatically performed for all amino acid residues in the peptide sequence. JR was synthesized on Fmoc-Rink Amide ProTide™ LL resin (0.20 meq/g substitution) or Fmoc-Rink Amide MBHA PS resin (0.33 meq/g substitution). [65-74]ACP, liraglutide, and semaglutide were synthesized on Fmoc-Gly-Wang-ProTide resin (0.24 meq/g substitution) and [1-42]β-Amyloid was synthesized on Fmoc-Ala-Wang-ProTide resin (0.23 meq/g substitution). Method details with stepwise operations for wash-free synthesis are described in Supplementary Table 3.

### Wash-free production scale liraglutide synthesis
Liraglutide was synthesized at 25 mmol scale using Fmoc-Gly-Wang-ProTide resin (0.24 meq/g substitution) in a 3 L reaction vessel on the Liberty PRO microwave peptide synthesizer. Couplings were performed for 5 min at 80 °C using Fmoc-amino acid (200 mL, 0.5 M in DMF), DIC (50 mL, 4 M in DMF), and Oxyma (225 mL, 0.33 M in DMF). After draining the post-coupling mixture, Fmoc deprotection step was performed for 10 min at 90 °C by adding 50 mL of pyrrolidine/DMF (15% v/v) followed by additional DMF (250 mL) to obtain a final concentration of 2.5% pyrrolidine in the reaction vessel. Nitrogen flow rate

at 85 L/min was directed through a spray head in the top of the reaction vessel to facilitate directed flushing of the headspace gas during each deprotection step. Fmoc-Lys(palmitoyl-Glu-OtBu)-OH was coupled using 2 equivalent excess with a wash-based coupling cycle, while all other amino acid residues in the sequence used no washings after the deprotection and coupling steps. Fmoc-His(Boc)-OH was coupled by using 2 x 30 min at 40 °C method. Wash-based cycles at 25 mmol scale used 4 x 650 mL DMF and 1 x 800 mL DMF for post-deprotection washings and Fmoc deprotection step was performed for 4 min at 80 °C. The cycles involving deprotection-coupling (for wash-free) or deprotection-washing-coupling (for wash-based) runs were automatically performed for all amino acid residues in the peptide sequence. Method details with stepwise operations for wash-free production scale synthesis are described in Supplementary Table 4.

### Protein synthesis
Proteins were synthesized using automated microwave synthesis conditions on a CEM Liberty PRIME 2.0 system at 0.10 mmol scale using the one-pot coupling/deprotection methodology[20,21]. Couplings were performed with Fmoc-amino acid (2.0 mL, 0.5 M in DMF), DIC (1.0 mL, 2.0 M in DMF) and Oxyma (1.75 mL, 0.50 M in DMF) for 30 seconds at room temperature followed by 4 min at 90 °C. Fmoc deprotection step was performed for 2 min at 110 °C and initiated by adding 0.75 mL of pyrrolidine/DMF (28% v/v) directly to the undrained post-coupling solution. Nitrogen headspace flushing (3.5 L/min) was used during the deprotection step. The wash-based method used 3 x 4 mL DMF post-deprotection washings. The cycles involving deprotection-coupling (for wash-free) or deprotection-washing-coupling (for wash-based) runs were automatically performed for all amino acid residues in the peptide sequence. Proinsulin 86-mer and Barstar 89-mer proteins were synthesized on Fmoc-Rink Amide ProTide™ LL resin (0.18 meq/g substitution).

### Resin cleavage (peptides)
The peptidyl resin was washed with DCM (3 x 15 mL) after synthesis. Cleavage was performed for 30 min at 38 °C using 5 mL of a freshly prepared cleavage cocktail [TFA/TIS/H$_2$O/DODT (92.5/2.5/2.5/2.5)]. The TFA solution was collected by filtration and ice-cold ethyl ether was added followed by centrifugation at 2,129 g for 3 min to obtain the crude peptide as a white pellet.

### Resin cleavage (proteins)
The peptidyl resin was washed with DCM (3 x 15 mL) after synthesis. Cleavage was performed for 5 h at RT using a slow cleavage method by adding 7.5 mL of [TFA/TIS/H$_2$O/DODT (6/0.5/0.5/0.5)] followed by a gradual addition of 4 mL TFA every hour for 3 hours. After the third addition (final conc. TFA/TIS/H$_2$O/DODT (18/0.5/0.5/0.5) the cleavage was allowed to react for an additional 2 hours. The TFA solution was collected by filtration and ice-cold ethyl ether was added followed by centrifugation at 2,129 g for 3 min to obtain the crude protein as a white pellet.

### Analysis
All peptides were lyophilized overnight after dissolving the pellet in 10% acetic acid/deionized water. A lyophilized aliquot of the peptide was taken in deionized water (~2 mg/mL peptide concentration) and a clear solution was obtained by the addition of acetonitrile, ammonium hydroxide (up to 1%), or acetic acid (up to 9%) followed by sonication. Protein samples (barstar and proinsulin) were dissolved by sonicating in a solution of H$_2$O/ACN/AcOH (8:1:1) for 1 h. The peptide/protein solution were analyzed on a Vanquish UHPLC system (Thermo Fisher; Waltham, MA, USA) with a Waters ACQUITY UPLC BEH C8 reversed-phase column (100 x 2.1 mm i.d., 1.7 μm, 130 Å; Waters Corporation, Milford, MA, USA) coupled to an Exactive™ Plus Orbitrap™ mass spectrometer (Thermo Fisher; Waltham,

MA, USA) via an ESI source (operated in positive polarity mode). Deconvoluted mass spectra for proteins were obtained using UniDec (Universal Deconvolution) Version 6.0.1 developed by Marty et al[49]. Analytical runs were performed at a flow rate of 0.5 mL/min with gradient elution of 10–70 % B using 0.05% trifluoroacetic acid in water (A) and 0.05% trifluoroacetic acid in acetonitrile (B). The column and autosampler were maintained at 40 and 24 °C, respectively for all peptides except $^{1-42}$β-amyloid. $^{1-42}$β-amyloid was analyzed on a Waters ACQUITY UPLC BEH C8 reversed-phase column (100 x 2.1 mm i.d., 1.7 μm, 130 Å; Waters Corporation, Milford, MA, USA) with a flow rate of 0.6 mL/min at 70 °C column temperature on a Waters Acquity RP-UPLC system with PDA detector coupled to a 3100 Single Quad mass spectrometer.

### Purification
Lyophilized protein samples (barstar and proinsulin) were dissolved (barstar: 6.4 mg/mL in water with 0.2 % ammonium hydroxide and 10 mM DTT; proinsulin: 8 mg/mL in 6 M GdnHCl with 0.1% ammonium hydroxide and 100 mM DTT through sonication for 1 h at 40 °C. Lyophilized liraglutide was dissolved (8.1 mg/mL) in 20 % acetonitrile. Samples were filtered with a 0.45 μm regenerated cellulose syringe filter (Phenomenex; Torrance, CA, USA) prior to purification. Purifications were completed on a CEM Prodigy HPLC System, which includes an integrated heating system (column oven and mobile phase heater) to enable high-efficiency elevated temperature operation. Barstar and proinsulin purifications were performed at 60 °C using a Waters Protein XBridge C4 column (19 x 150 mm, 5 μm, 300 Å; Waters Corporation, Milford, MA, USA) with mobile phases consisting of 0.1 % trifluoroacetic acid in water (A) and 0.1 % trifluoroacetic acid in acetonitrile (B). Liraglutide purifications were performed using the same conditions, but with a Waters XBridge C8 column (19 x 150 mm, 5 μm, 130 Å). Optimized gradient conditions were determined by first injecting ~10 mg of crude sample on a 10–70 % B screening gradient (20 min gradient; ~3 % B/CV) with a flow rate of 27 mL/min. The target peak retention time was then used to calculate (using CEM Focused Gradient Calculator software, version 1.1.673.1159) optimized focused gradients for each purification. The protein samples were purified using focused gradients over 25 min (proinsulin: 27–39 % B; barstar: 39–51 % B), while liraglutide was purified using a focused gradient over 18 min. (46–55 % B).

### GC-FID analysis
Each sample was diluted 100-fold by serial dilution with HPLC-grade isopropyl alcohol (IPA) (1:10, repeated twice) and then injected into a Shimadzu Nexis GC-2030 equipped with an SH-I-5Sil MS 1,4-bis(dimethylsiloxy)phenylene dimethyl polysiloxane column (0.25 mm ID x 30.0 m, 0.25 μm film thickness) and FID-2030 detector (Shimadzu Scientific Instruments, Inc.; Columbia, MD, USA). Injections were performed with an injection volume of 1.0 μL with a split ratio of 10.0 into an injection port at 250 °C. The column oven was heated from 80 °C to 200 °C with a 1 min hold at 80 °C, followed by a 10 °C/min ramp, and a 1 min hold at 200 °C. The concentration of pyrrolidine was determined by first preparing and analyzing a series of pyrrolidine standards: 5%, 1%, 0.5%, 0.05% (v/v in DMF). Each standard was diluted 100-fold by serial dilution with HPLC-grade IPA (1:10, repeated twice), and then injected onto the GC-FID with the method above for analysis. The pyrrolidine peak areas for the standards were fitted by linear regression using LabSolutions software to give a standard calibration curve with an equation of $f(x) = 83769.2 \cdot x - 2634.55$ and $R^2$ value of 0.9995893. The equation was saved in LabSolutions and used to automatically calculate the concentration of pyrrolidine from the pyrrolidine peak area for the residual sample analysis.

## Data availability
All of the relevant data is available in Supplementary Information.

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

## Acknowledgements

We would like to thank David Herman, Will Sweatman, Tony Baranski, and Joshua Foster of CEM Corporation for their support with the development of hardware and software components of the automated synthesizers. Lee Estep and Scott Rifenburgh are acknowledged for artistic design of the illustrations. We thank Benedict Liu for his assistance with GC-FID analysis. We gratefully acknowledge Dr. Michael J. Collins for discussions and continued encouragement throughout this project.

## Author contributions

T.A.W. conducted experiments for wash-free method development for peptides. D.J.C. performed the synthesis and purification of proteins. C.L.S. performed the purification of peptides and proteins. L.J.T. developed instrumentation for the automated synthesizers. C.L.H. assisted with the large-scale peptide synthesis. J.M.C. and S.K.S. conceptualized the project and wrote the paper with the other authors.

## Competing interests

The authors declare the following competing interests: All authors work at CEM Corporation which develops and manufactures the peptide synthesizers used in this study. J.M.C. and S.K.S. are co-inventors on a U.S. provisional patent application with application serial number 63/401,349 filed by CEM Corporation. The invention covers the aspects of headspace gas flushing for wash-free solid-phase peptide synthesis.
