## [Peer Review File · Nature Communications]

Reviewers' Comments:

Reviewer #1:

Remarks to the Author:

The authors report in this manuscript an advanced protocol for the solid phase synthesis of peptides, claimed to be green and sustainable. Increasing the greenness of peptide synthesis is a hot topic faced in the scientific community by replacing toxic solvents and problematic reagents with more sustainable ones, or reducing the volume of them, without considering the issues related to their nature. This work follows the latter approach, since reprotoxic DMF is used, but washings are avoided both in coupling and deprotection steps. The manuscript reports the successful application of the protocol to several therapeutic peptides, but the approach has been tested mainly with DMF (only NBP has been tested as greener alternative) and only with pyrrolidine as the deprotecting base. The scope of the study has been investigated by synthesizing several different peptides but has not been extended to all the solvents, all the coupling reagents and all the possible bases, thus being, at this stage, only a technological advancement applied to the conditions commonly used in microwave SPPS synthesizers. The interest for the protocol would be much higher if applied to other SPPS conditions and technologies.

As it is now, the manuscript is not suitable for Nature Communications and it's more suitable to a more specific audience as the ones of Journal of Peptide Science or Organic Process Research and Development.

Some suggestions for revisions:

- In figure 1 all the products and coproducts of the synthesis are reported but it's not discussed their fate in any part of the text. Since the optimized protocol avoids any washing of the resin during peptide growth, it's quite clear that part of the coproducts will stay inside the reactor and will be completely removed only after cleavage and precipitation of the peptide. This issue should be underlined and possible side reactions discussed.
- In table 1, when crude purity is very low (entries 3 and 5 for instance) the other components of the mixture should be disclosed
- In figure 2, the detector of the chromatograms (MS? UV? Which wavelenght?) has to be described in the figure caption
- Figure 3, liraglutide synthesis – 25 mmol scale: the purity of the wash-free method seems to be better than that of the wash-method, in Table 1 the purity value for the wash-free synthesis is not reported, details should be complete and explained
- In the supporting material file, Figure 3 and 4 as well as the pictures of the synthesizers are not necessary and can be removed.

Reviewer #2:

Remarks to the Author:

Solid-phase peptide synthesis (SPPS) is widely used for the small to industrial scale synthesis due to its simple protocol, which enables the automated synthesis. One of the major drawback of the SPPS is that it produces large amount of waste solvents especially in the case of larger scale synthesis. The authors group previously succeeded to omit the washing step after the coupling reaction and reduced the amount of the solvent used for the washing (typically, DMF). However, the washing after the deprotection of the α -amino group by piperidine etc., could not be avoided, since the remaining piperidine causes side reactions in the subsequent amino acid introduction reaction, such as the removal of the α -amino protecting group of the amino acid and quench of the activated amino acid by the nucleophilic attack.

In this paper, they cleverly overcame the problem by using reduced amount of the amino-group deblocking reagent with the lower boiling point than piperidine, pyrrolidine. It was directly introduced to the solution of amino acid coupling mixture at 110 °C, which is much higher than the boiling point of pyrrolidine. With the assistance of nitrogen gas bubbling from the bottom and flashing at the top of the reaction vessel, the excess amount of the pyrrolidine was effectively removed by evaporation during the deprotection step, which resulted in the omission of the washing step after the deblocking reaction. By this method, the reduction of the waste of the reaction was up to 95% of the usual method, which makes the SPPS more environmentally friendly and cost effective. Considering the growing interest in the peptide therapeutics, this reviewer thinks that the paper is suitable for publication in Nat. Commun. with minor modifications. Please

check the following points.

1. In the production scale synthesis of liraglutide by wash free method, why was the amino acid coupling mixture first removed and then the pyrrolidine/DMF added?
2. The amount of Oxyma is less than the Fmoc amino acid. Simply thinking, all the amino acid is better to be activated at the same time to reduce the activation-coupling reaction. If there is a particular reason to reduce the amount, it is better to add the description.
3. If possible, the amount of the remaining pyrrolidine in the resin after draining the deprotective solvent is better to be checked.
4. Asp-Gly sequence is particularly prone to form succinimide side product. How much amount of the side reaction occur, when this method is used? Please add comments, if possible.

Reviewer #3:

Remarks to the Author:

The article describes a new approach to Solid Phase Peptide Synthesis that has the potential to reduce the Process Mass Index of chemical synthesis of peptides, by removing the washes used post deprotection to remove the secondary amine required in Fmoc synthesis. Due to the increase demand for large amount of Peptide API for Weight Loss application, these efforts are relevant and make the article in scope of Nature Communication. This is well described in the introduction (although in the introduction few items are not correctly reported, since there are no Video Files in the supplementary folder, or the number of washes is reported as a factor driving a low PMI, while is exactly the opposite).

Conceptually the approach described (evaporation of the base using microwave under nitrogen stream), allows to reuse the coupling solution by adding the deprotection agent to the reactor (telescoping the reaction), and eliminating it via evaporation, could reduce the need for washes post deprotection, could strongly reduce the waste of significant amount of solvents, and decrease the PMI of SPPS. This concept is extremely interesting and is in the scope of the Nature Communication.

On the other hand the article fails to describe accurately in detail the process and to provide few key data that would support the actual fundamental execution of the work and to support the main claim (reduction of pyrrolidine concentration via evaporation).

For the article to be acceptable for publication we suggest to execute the following studies:

1) Since no data re provided quantifying the level of pyrrolidine before and after the deprotection step (prior to the addition of the following amino acid solutions) the level of pyrrolidine in the reactor at different steps of the deprotection should be established (for example using Head Space GC).

2) Since the author report the scalability of the process from 0.1 mmol to 25 mmol we suggest that the pyrrolidine quantification pre and post deprotection /evaporation is establish for both instruments and compared.

For the article to be accepted, more detailed information about program used for the syntheses both at 0.1 mmol scale and 25 mmol scale should be included. As an example this reviewer did not understand where and if the solutions were drained from the reactor, or how the amino acids and coupling reagent were introduced (as concentrated bulk into the post deprotection reaction medium or as fresh solution). Based on the described process it would seem that the initial solution used for the first derivative were kept for all the following coupling of each amino acid, which seems unlikely.

It is common for article describing automated SPPS to provide the program for each cycle in a table describing each step that the synthesizer execute (as example "add pyrrolidine", "nitrogen from the reactor bottom at xxl/min flow", "drain the reactor", and so on). With this information, any scientist could try to reproduce this approach, and adapting their equipment to achieve similar results. As it stands now, the screen shot of the software in the Supplementary information does not provide enough information.

Throughout the article, the Nitrogen gas pressure is provided without specifying for which instrument this refer to. At one point nitrogen is reported as 85 psi, while in another section is reported as "85 L/min nitrogen pressure". This reviewer think that rather than providing the gas pressure, in this specific case the gas flow at a specific step is a key parameter affecting the evaporation of the pyrrolidine. Therefore, we suggest that the actual gas flow is established using a flow meter of the vent line of the reactor. It would be important to report the actual nitrogen

flow both for the 0.1 mmol and the 25 mmol processes, since these two flow should correlate with the efficiency of the reduction of pyrrolidine that will be established by the quantitative analysis. We suggest checking the reference to the supplementary information in the main text. In the introduction there is a reference to a video in the supplementary information, which was not present in the data provided in the review process, nor is present in the Supplementary Information index.

In our opinion, with the additional quantitative data on pyrrolidine level, and actual nitrogen flow during evaporation , and with a more detailed description of the steps executed by the synthesizer, the article could be consider for publication.

Total Wash Elimination for Solid Phase Peptide Synthesis

RESPONSE TO REVIEWER COMMENTS

The original comments, suggestions and questions from the reviewers are followed by our point-by-point responses (in blue). All new additions and changes in the revised manuscript and supplementary information files are shown with color highlighting.

Reviewer #1 (Remarks to the Author):

The authors report in this manuscript an advanced protocol for the solid phase synthesis of peptides, claimed to be green and sustainable. Increasing the greenness of peptide synthesis is a hot topic faced in the scientific community by replacing toxic solvents and problematic reagents with more sustainable ones, or reducing the volume of them, without considering the issues related to their nature. This work follows the latter approach, since reprotoxic DMF is used, but washings are avoided both in coupling and deprotection steps. The manuscript reports the successful application of the protocol to several therapeutic peptides, but the approach has been tested mainly with DMF (only NBP has been tested as greener alternative) and only with pyrrolidine as the deprotecting base. The scope of the study has been investigated by synthesizing several different peptides but has not been extended to all the solvents, all the coupling reagents and all the possible bases, thus being, at this stage, only a technological advancement applied to the conditions commonly used in microwave SPPS synthesizers. The interest for the protocol would be much higher if applied to other SPPS conditions and technologies.

As it is now, the manuscript is not suitable for Nature Communications and it's more suitable to a more specific audience as the ones of Journal of Peptide Science or Organic Process Research and Development.

Response:

We thank the reviewer's valuable comments and suggestions for increasing the scope of this methodology by testing other solvents, coupling reagents and bases. An inherent feature of this new wash free SPPS process is the use of elevated temperature reaction conditions. As such a higher boiling point solvent is important so that the deprotection base evaporates from the system before the solvent. We expanded the list of higher boiling point solvents tested to include NMP in addition to NBP and DMF that were previously described. These new results have been added in the revised manuscript (Table 1).

Similarly, we have also added new results obtained by testing the traditionally used piperidine as a base. We agree that adding baseline results with the traditionally used piperidine would be useful even with the preferred characteristics of pyrrolidine. This has been done in Table 1 and demonstrated that piperidine even with a higher boiling point (106 °C vs. 87 °C) can be

successfully utilized. However, pyrrolidine appears preferable due to its (a) classification as a non-restricted chemical (unlike piperidine), (b) ability to achieve more complete deprotection, and (c) its similar pricing and availability to piperidine.

In regards to testing other coupling reagents, previous studies^{18,19} established that carbodiimide based activation is fundamentally preferable over onium salt-based activation (HBTU, HATU, PyBOP, PyAOP, etc.) particularly with elevated temperature reaction conditions. This is due to the requirement of onium salt-based activation to need a relatively strong base diisopropylethylamine (DIEA) to facilitate activation. The presence of DIEA is directly responsible for side-reactions (ex. epimerization) that are exacerbated at elevated temperature.

It is well known that the presence of base (DIEA) during the activation and coupling causes side-reactions that are made worse with increasing temperature. In particular, DIEA is directly responsible for epimerization which limits the use of onium salts under cGMP processes even at room temperature conditions. For this reason, the most common and widely used activation method with both traditional and microwave SPPS process is based on the use of carbodiimides. More aggressive activation strategies such as onium salts are generally used at R&D scales under non-GMP conditions with room temperature conditions. They are also explored in R&D as a means to enhance coupling in combination with greener solvents that may reduce coupling efficiency.

The reviewers' comments also guided us to provide some brief discussion in the text to explain factors that influence the choice of reagents and solvents. We added text on Page 8 to discuss the results obtained with NMP and piperidine and the need of alternative solvents to have a significantly higher boiling point than the deprotection base used. This precludes the use of some recently explored solvents for SPPS such as THF, Me-THF, and Ethyl Acetate.

Text was also incorporated to provide the reader with context of carbodiimide activation as not only a standard activation method (used with both non-microwave and microwave peptide synthesis), but also one that avoids known problems from using other activation approaches under elevated temperature conditions.

“The methodology was designed using traditional carbodiimide based activation with N,N'-diisopropylcarbodiimide (DIC) and ethyl 2-cyano-2-(hydroxyimino)acetate (Oxyma Pure). Unlike phosphonium and aminium-based coupling, this activation strategy is extraordinarily tolerant to elevated temperatures because it avoids the use of excess base, such as DIEA, which leads to epimerization and other problematic side reactions when heated¹⁹.”

Thus, we believe carbodiimide chemistry is not only the most suitable activation strategy for elevated temperature conditions in use with the wash-free methodology, but also a very general and widely-used activation methodology that allows utilization in the synthesis of all types of peptides and proteins with high crude purities and minimal epimerization as shown in the present work (and prior references, ex. #19). Accordingly, at this point we have not made any further attempts to test other coupling reagents.

Some suggestions for revisions:

- In figure 1 all the products and coproducts of the synthesis are reported but it's not discussed their fate in any part of the text. Since the optimized protocol avoids any washing of the resin during peptide growth, it's quite clear that part of the coproducts will stay inside the reactor and will be completely removed only after cleavage and precipitation of the peptide. This issue should be underlined and possible side reactions discussed.

Response:

We appreciate the reviewer for this important suggestion. We have now added a detailed discussion on Page 4 of the revised manuscript explaining the formation of all coproducts and the final outcome at the end of coupling and deprotection. We have also added full method details describing the stepwise operations used for 0.1 and 25 mmol runs in Supplementary Tables 3 and 4, respectively.

- In table 1, when crude purity is very low (entries 3 and 5 for instance) the other components of the mixture should be disclosed

Response:

On Page 7, we have added a sentence identifying the deletion side products for entry 3. Side products formed in entry 5 are also mentioned in the same paragraph.

- In figure 2, the detector of the chromatograms (MS? UV? Which wavelength?) has to be described in the figure caption

Response:

We have added the UV wavelength information in the captions of Figures 3 and 4. MS details are included in the Analysis section on Page 19.

- Figure 3, liraglutide synthesis – 25 mmol scale: the purity of the wash-free method seems to be better than that of the wash-method, in Table 1 the purity value for the wash-free synthesis is not reported, details should be complete and explained

Response:

We thank the reviewer for this careful observation. UPLC-MS purity reports obtained by integration of all peaks indicate very similar crude purities for wash-free (77%) and wash-based (78%) methods. We re-ran UPLC-MS analysis of the wash-based sample with very careful attention to identical sample preparation. The updated chromatogram has now replaced the previous version and shows a very similar trace as compared to the wash-free sample. We

have also rearranged Supplementary Table 1 to show the actual purity values in the same row for easy comparison of wash-free and wash-based synthesis of Liraglutide.

- In the supporting material file, Figure 3 and 4 as well as the pictures of the synthesizers are not necessary and can be removed.

Response:

We appreciate this feedback and addressed this by removing Figures 3 and 4 as well as all pictures and videos of the synthesizers.

Reviewer #2 (Remarks to the Author):

Solid-phase peptide synthesis (SPPS) is widely used for the small to industrial scale synthesis due to its simple protocol, which enables the automated synthesis. One of the major drawback of the SPPS is that it produces large amount of waste solvents especially in the case of larger scale synthesis. The authors group previously succeeded to omit the washing step after the coupling reaction and reduced the amount of the solvent used for the washing (typically, DMF). However, the washing after the deprotection of the α -amino group by piperidine etc., could not be avoided, since the remaining piperidine causes side reactions in the subsequent amino acid introduction reaction, such as the removal of the α -amino protecting group of the amino acid and quench of the activated amino acid by the nucleophilic attack.

In this paper, they cleverly overcame the problem by using reduced amount of the amino-group deblocking reagent with the lower boiling point than piperidine, pyrrolidine. It was directly introduced to the solution of amino acid coupling mixture at 110 °C, which is much higher than the boiling point of pyrrolidine. With the assistance of nitrogen gas bubbling from the bottom and flashing at the top of the reaction vessel, the excess amount of the pyrrolidine was effectively removed by evaporation during the deprotection step, which resulted in the omission of the washing step after the deblocking reaction. By this method, the reduction of the waste of the reaction was up to 95% of the usual method, which makes the SPPS more environmentally friendly and cost effective. Considering the growing interest in the peptide therapeutics, this reviewer thinks that the paper is suitable for publication in Nat. Commun. with minor modifications. Please check the following points.

Response:

We appreciate the reviewer for their encouraging comments summarizing the value of wash-free methodology.

1. In the production scale synthesis of liraglutide by wash free method, why was the amino acid coupling mixture first removed and then the pyrrolidine/DMF added?

Response:

We thank the reviewer for raising this important question. The reason for this change in operation was due to current hardware limitations with the large-scale synthesizer. The main purpose of the production scale run was to establish that the concept of wash-free synthesis works not only for research scale but for production scale as well. We plan to explore the use of non-drained coupling at this scale with a future upgrade to the system that will allow for addition of concentrated deprotection through a dedicated path.

2. The amount of Oxyma is less than the Fmoc amino acid. Simply thinking, all the amino acid is better to be activated at the same time to reduce the activation-coupling reaction. If there is a particular reason to reduce the amount, it is better to add the description.

Response:

We thank the reviewer for this comment. During our optimization of wash-free methodology, we recognized that up to one-equivalent less of Oxyma could potentially be used during the coupling reaction. The reason for this is that a by-product of the acylation step is the release of a free Oxyma. Therefore, the Oxyma freed from acylation would then immediately be available to react with excess Fmoc amino acid and DIC and thereby be sequestered without reducing the available activated species. Using one equivalent less of Oxyma is beneficial for overall efficiency while providing theoretical benefits (protection against acid sensitive side reactions, protection against hydrogen cyanide formation from known reaction of DIC and Oxyma). We have included this detail in the new discussion added on Page 6 of the revised manuscript.

3. If possible, the amount of the remaining pyrrolidine in the resin after draining the deprotective solvent is better to be checked.

Response:

We appreciate this important suggestion from the reviewer. We performed GC analysis to find out the amount of remaining pyrrolidine after draining at the end of deprotection step and we have now added these results on Page 15 of the revised manuscript.

4. Asp-Gly sequence is particularly prone to form succinimide side product. How much amount of the side reaction occur, when this method is used? Please add comments, if possible.

Response:

Asp-Gly sequence is particularly prone to form aspartimide/succinimide side product, and the use of HOBt or Oxyma additive with the deprotection base is a well-known strategy to reduce the formation of this side product in SPPS. As shown in Figure 2, the deprotection step in one-pot wash-free methodology has an added advantage of inherent presence of Oxyma from the undrained post-coupling mixture. Due to this, we have generally noticed a reduction in aspartimide formation but an extensive study has not been undertaken to quantify the actual levels. We recommend use

of Fmoc-Asp(OMpe)-OH for Asp residues and Fmoc-Asp(OtBu)-(Dmb)Gly-OH for Asp-Gly to minimize aspartimide formation.

Reviewer #3 (Remarks to the Author):

The article describes a new approach to Solid Phase Peptide Synthesis that has the potential to reduce the Process Mass Index of chemical synthesis of peptides, by removing the washes used post deprotection to remove the secondary amine required in Fmoc synthesis. Due to the increase demand for large amount of Peptide API for Weight Loss application, these efforts are relevant and make the article in scope of Nature Communication. This is well described in the introduction (although in the introduction few items are not correctly reported, since there are no Video Files in the supplementary folder, or the number of washes is reported as a factor driving a low PMI, while is exactly the opposite).

Conceptually the approach described (evaporation of the base using microwave under nitrogen stream), allows to reuse the coupling solution by adding the deprotection agent to the reactor (telescoping the reaction), and eliminating it via evaporation, could reduce the need for washes post deprotection, could strongly reduce the waste of significant amount of solvents, and decrease the PMI of SPPS. This concept is extremely interesting and is in the scope of the Nature Communication.

On the other hand the article fails to describe accurately in detail the process and to provide few key data that would support the actual fundamental execution of the work and to support the main claim (reduction of pyrrolidine concentration via evaporation).

Response:

We appreciate the reviewer for their encouraging comments. We have now provided additional data and method details with stepwise operations in the revised manuscript and supplementary information files as described in the following responses.

For the article to be acceptable for publication we suggest to execute the following studies:

- 1) Since no data re provided quantifying the level of pyrrolidine before and after the deprotection step (prior to the addition of the following amino acid solutions) the level of pyrrolidine in the reactor at different steps of the deprotection should be established (for example using Head Space GC).
- 2) Since the author report the scalability of the process from 0.1 mmol to 25 mmol we suggest that the pyrrolidine quantification pre and post deprotection /evaporation is establish for both instruments and compared.

Response (1 + 2):

We appreciate this important suggestion from the reviewer. We performed GC analysis from the 0.1mmol scale process to find out the residual level of pyrrolidine in the subsequent coupling and we have now added these results on Page 15 of the revised manuscript. We have also compared

the levels measured to other references (ref. # 45, 46) that detailed residual base measured in SPPS processes. This provides a general guideline of ideally achieving < 2,000 ppm for residual pyrrolidine for larger scales.

The purity and epimerization results from the larger 25mmol experiment were similar to the 0.1mmol scale process. This indicates that the residual pyrrolidine level under the 25mmol conditions isn't significantly higher than the 0.1mmol process (which was measured at < 2,000 ppm). We therefore haven't attempted to re-run the larger scale experiment to obtain exact residual pyrrolidine content due to cost burden from re-running at this scale.

For the article to be accepted, more detailed information about program used for the syntheses both at 0.1 mmol scale and 25 mmol scale should be included. As an example this reviewer did not understand where and if the solutions were drained from the reactor, or how the amino acids and coupling reagent were introduced (as concentrated bulk into the post deprotection reaction medium or as fresh solution). Based on the described process it would seem that the initial solution used for the first derivative were kept for all the following coupling of each amino acid, which seems unlikely.

It is common for article describing automated SPPS to provide the program for each cycle in a table describing each step that the synthesizer execute (as example "add pyrrolidine", "nitrogen from the reactor bottom at xxl/min flow", "drain the reactor", and so on). With this information, any scientist could try to reproduce this approach, and adapting their equipment to achieve similar results. As it stands now, the screen shot of the software in the Supplementary information does not provide enough information.

Throughout the article, the Nitrogen gas pressure is provided without specifying for which instrument this refer to. At one point nitrogen is reported as 85 psi, while in another section is reported as "85 L/min nitrogen pressure". This reviewer think that rather than providing the gas pressure, in this specific case the gas flow at a specific step is a key parameter affecting the evaporation of the pyrrolidine. Therefore, we suggest that the actual gas flow is established using a flow meter of the vent line of the reactor. It would be important to report the actual nitrogen flow both for the 0.1 mmol and the 25 mmol processes, since these two flow should correlate with the efficiency of the reduction of pyrrolidine that will be established by the quantitative analysis.

Response:

We thank the reviewer for providing important suggestions to add method details for 0.1 and 25 mmol syntheses as well as the nitrogen flow parameters. As suggested, we have now added full method details describing the stepwise operations used for 0.1 and 25 mmol runs in Supplementary Tables 3 and 4, respectively. In the Methods section for research scale and production scale syntheses on Page 17, we have now added a sentence with reference to Supplementary Tables 3 and 4 for obtaining further details and stepwise operations. As suggested, we have also added the values for actual gas flow (L/min) using a flow meter connected to the reactors for the 0.1 mmol and the 25 mmol processes. We thank the reviewer for pointing out the typing mistake in reporting 85 psi instead of 85 L/min and this has been corrected in the revised manuscript.

We suggest checking the reference to the supplementary information in the main text. In the introduction there is a reference to a video in the supplementary information, which was not present in the data provided in the review process, nor is present in the Supplementary Information index.

Response:

We thank the reviewer for pointing this out. In consideration of comments from other reviewers, we have now removed all of the pictures and videos of the synthesizers.

In our opinion, with the additional quantitative data on pyrrolidine level, and actual nitrogen flow during evaporation, and with a more detailed description of the steps executed by the synthesizer, the article could be consider for publication.

Response:

We appreciate the reviewer for these valuable suggestions and we have now provided all of the additional data and detailed descriptions in the revised manuscript and supplementary information files.

Reviewers' Comments:

Reviewer #1:

Remarks to the Author:

The authors correctly addressed all the referees' requests, increasing the quality of the paper that is now ready to be accepted for publication

Reviewer #2:

Remarks to the Author:

In the revised manuscript, the questions raised by the reviewer was appropriately responded. I think that the paper is now suitable for publication in the present form.

Reviewer #3:

Remarks to the Author:

The authors addressed all the objections and request for data I raised. The new data are in line with the expectation, and strengthen and confirms the author hypothesis.

In addition I carefully reviewed the comments from the other reviewers, and I think the authors addressed them in a satisfactory way. This new data and the corrections resulting from the other reviewers' comments made the article complete and ALMOST ready for publication.

One comment I made (maybe I was not clear enough in the initial review) still remain unchanged, and should be changed before publication. Specifically in the introduction, in this paragraph: "Historically, about 5 washes have been needed between each step resulting in a large majority of the total waste generated and driving a low process mass intensity (PMI)."

the word LOW should be replaced by HIGH.

A low PMI is indicative of an efficient process, and a High PMI is indicative of a wasteful process. If the waste increases as described in the paragraph due to many washes, the PMI increases.

Once this misunderstanding is corrected, in my opinion the article is ready for publication.

Total Wash Elimination for Solid Phase Peptide Synthesis

RESPONSE TO REVIEWER COMMENTS

The original comments, suggestions and questions from the reviewers are followed by our point-by-point responses (in blue).

REVIEWERS' COMMENTS

Reviewer #1 (Remarks to the Author):

The authors correctly addressed all the referees' requests, increasing the quality of the paper that is now ready to be accepted for publication

Reviewer #2 (Remarks to the Author):

In the revised manuscript, the questions raised by the reviewer was appropriately responded. I think that the paper is now suitable for publication in the present form.

Reviewer #3 (Remarks to the Author):

The authors addressed all the objections and request for data I raised. The new data are in line with the expectation, and strengthen and confirms the author hypothesis.

In addition I carefully reviewed the comments from the other reviewers, and I think the authors addressed them in a satisfactory way. This new data and the corrections resulting from the other reviewers' comments made the article complete and ALMOST ready for publication.

One comment I made (maybe I was not clear enough in the initial review) still remain unchanged, and should be changed before publication. Specifically in the introduction, in this paragraph: "Historically, about 5 washes have been needed between each step resulting in a large majority of the total waste generated and driving a low process mass intensity (PMI)."

the word LOW should be replaced by HIGH.

A low PMI is indicative of an efficient process, and a High PMI is indicative of a wasteful process. If the waste increases as described in the paragraph due to many washes, the PMI increases.

Once this misunderstanding is corrected, in my opinion the article is ready for publication.

Response:

We thank the reviewer #3 for catching this error. As suggested, we have now replaced the word "low" by "high" in the Introduction section.